# Distributed Unlearning with Lossy Compression

## Abstract

Machine unlearning enables to remove the contribution of a set of data points from a trained model. In a distributed setting, where a server orchestrates training using data available at a set of remote users, unlearning is essential to cope with the possible presence of malicious users. Existing distributed unlearning algorithms require the server to store all model updates observed in training, leading to immense storage overhead for preserving the ability to unlearn. In this work we study lossy compression schemes for facilitating distributed server-side unlearning with limited memory footprint. We identify suitable lossy compression mechanisms based on random lattice coding and sparsification. For a family of stochastic compression schemes encompassing probabilistic and subtractive dithered quantization, we derive an upper bound on the difference between the desired model that is trained from scratch and the model unlearned from lossy compressed stored updates. Our bound outperforms the state-of-the-art known bounds for non-compressed decentralized server-side unlearning, even when lossy compression is incorporated. We further provide a numerical study, shows that suited lossy compression can enable distributed unlearning with notably reduced memory footprint at the server while preserving the utility of the unlearned model.

## 1 Introduction

Deep learning usually requires large volumes of training data to result with high-performance models. While data is often abundantly available in the 'big data' era Jordan & Mitchell (2015), the source of the data might raise privacy or ownership concerns Liu et al. (2021), among which is the GDPR right to be forgotten (RTBF) Voigt & Von dem Bussche (2017), as well as security concerns, as adversaries can maliciously modify the training (poisoning or backdoors attacks) or test data (evasion attacks) Pitropakis et al. (2019). To address those, the paradigm of *machine unlearning* aims to selectively remove the influence of certain data points from a trained model, with neither retraining it from scratch nor impacting its original performance and predictive power Bourtoule et al. (2021).

In distributed learning systems, such as federated learning (FL) Zhang et al. (2021), where training is done on edge devices without data sharing, the ability to unlearn is often essential, particularly as users may be identified as adversaries after having already contributed to the learning procedure Nguyen et al. (2022). Nonetheless, the fact that the data whose contribution is to be removed is not often available makes unlearning in distributed settings more complex compared to conventional centralized setups Fraboni et al. (2024); Huynh et al. (2024); Tao et al. (2024). As a result, unlearning in distributed settings, e.g., federated unlearning (FU), relies on discarding historical parameter updates of the removed user. Accordingly, the server which orchestrates the learning procedure has to store all past contributions of all clients in order to have the ability to unlearn when required Gao et al. (2024). This induces a notable limitation, requiring excessive and possibly prohibitive overhead for storing a large number of highly parameterized updates for each user.

The goal of this work is to alleviate the storage burden introduced in server-side unlearning using *lossy source coding* tools. To do so, we have the server not saving the users' sent model updates, but alternatively a lossy compressed versions, notably reducing its memory footprint. We comprehensively explore lossy compression techniques, based on (probabilistic) lattice quantization and sparsifcation, identified as suitable for server-side unlearning. The effect of their induced distortion on the unlearned model is further analytically and experimentally analyzed using conventional metrics in unlearning literature. It is then revealed that while integrating proper compression into unlearning significantly relieve the server's storage load, it does not change the asymptotic behaviour

of the unlearned model, while effectively removing the influence of the unlearned user. Our main contributions are summarized as follows:

- We study lossy compression in a distributed (federated) unlearning framework, and identify the main considerations to mitigate the memory footprint accumulated over the learning procedure. We are, to the best of our knowledge, the first work that systematically examines the theoretical and numerical aspects of compressed distributed machine unlearning.

- We provide a theoretical analysis for gradient descent based distributed learning. We derive an upper bound on the commonly adopted proximity to the optimal train-from-scratch model of server-side unlearning, under a family of lossy compression mechanisms.

- We show that in the asymptotic regime, for growing amount of gradient descent iterations, if the learning rates gradually decay; then the compressed unlearned model does not diverge. Our characterization specializes also non-compressed distributed unlearning, and improves the state-of-the-art asymptotic behaviour of the bound presented in Huynh et al. (2024).

- All compression mechanisms are quantitatively validated in a FU experimental setup, evaluated using an established backdoor attack. There, it is demonstrated that lossy compression significantly lowers the memory footprint of unlearning, with only a minor degradation in accuracy and while preserving the ability to defend against malicious users attack.

## 2 RELATED WORK

**Server-Side Unlearning.** Distributed unlearning frameworks are categorized based on the identity of the users participating in the unlearning procedure Liu et al. (2023). *Server-side unlearning*, also termed *passive unlearning*, is the challenging setting in which only the server, who originally orchestrated the distributed learning procedure, participates in unlearning. Such scenarios correspond to, e.g., late detection of a malicious user, possibly after learning is concluded Wang et al. (2023).

Server-side unlearning typically depends on the utilization of stored historical data as gradients, global models, and contribution information. In FedRecovery Zhang et al. (2023), in addition to retaining the clients historical data, the server also quantifies their contributions based on gradient residuals. Upon an unlearning request, the server removes the unlearned user past contributions through a fine-tuning process. A more efficient version is then suggested by Crab Jiang et al. (2024), which uses only selective historical information and further assists a less-maliciously-degraded historical model than the initial one. The recovery process can be improved by introducing constraints, e.g., a penalty term based on projected gradients Fu et al. (2024); Shao et al. (2024); randomly initialized degradation models Zhao et al. (2023); estimated skew Huynh et al. (2024); and retraining based on the change of sampling probability Tao et al. (2024). In VeriFi Gao et al. (2024), the target client collaborates with the server and marks his data to verify the unlearning. As means to preserve performance despite the target client contribution elimination, knowledge distillation was shown to facilitate information transfer from the trained model to the unlearned one Wu et al. (2022; 2023).

**Distributed Learning with Lossy Compression.** While lossy compression techniques are still unexplored for distributed unlearning, various schemes have been considered for distributed training, particularly in the aim of alleviating uplink communication bottlenecks Chen et al. (2020); Li et al. (2020). Among which are sub-sampling or sparsification Lin et al. (2017); Hardy et al. (2017); Aji & Heafield (2017); Konečný et al. (2016); Stich et al. (2018); and probabilistic scalar Wen et al. (2017); Alistarh et al. (2017); Horvóth et al. (2022); Reisizadeh et al. (2020); Horváth et al. (2023) or vector quantization Lang et al. (2023a;b); Azimi-Abarghouyi & Varshney (2024). Lossy compression, as opposed to its lossless counterpart, inevitably induces distortion, yet enables substantial memory savings Polyanskiy & Wu (2014). Nevertheless, the random distortion induced by probabilistic lossy compression can be rendered to have a negligible effect on the learning procedure Alistarh et al. (2017); Shlezinger et al. (2020). Inspired by that, we seek for lossy compression methods that notably reduce memory footprint in server side unlearning while retaining model accuracy.

## 3 SYSTEM MODEL

**Distributed Learning.** We consider a server training a model with parameters $\boldsymbol{w} \in \mathbb{R}^m$ using data available at $U$ users, indexed by $u \in [U] := \{1, \ldots, U\}$. Let $\mathcal{L}_u(\boldsymbol{w})$ denote the $u$th

user empirical risk, the desired model is defined to be the minimizer of their average, that is $\arg\min_{\boldsymbol{w}} \left\{ \frac{1}{U} \sum_{u=1}^{U} \mathcal{L}_u(\boldsymbol{w}) \right\}$. Unlike conventional centralized learning, the datasets $\{\mathcal{D}_u\}$ are not shared with the server due to, e.g., privacy considerations, and thus the learning is federated Kairouz et al. (2021), operates in rounds. For every round $t$, the server distributes the global model $\boldsymbol{w}_t$ to the users. Each user locally performs several training iterations using its local $\mathcal{D}_u$, via, e.g., stochastic gradient descent (SGD), to update $\boldsymbol{w}_t$ into $\boldsymbol{w}_t^u$; and shares with the server the model update, i.e., $\boldsymbol{h}_t^u := \boldsymbol{w}_t^u - \boldsymbol{w}_t$. The server in turn collects the model updates from all participating users and aggregates McMahan et al. (2017) them into an updated global model via

$$\boldsymbol{w}_{t+1} := \boldsymbol{w}_t + \frac{1}{U} \sum_{u=1}^{U} \boldsymbol{h}_t^u, \tag{1}$$

where $\boldsymbol{w}_0$ is the vector of initial weights. For simplicity, equation 1 is formulated with all users participating in each round, which straightforwardly extends to partial user participation McMahan et al. (2017). The updated global model is again distributed to the users, and the learning procedure continues until convergence is reached. The above steps are summarized as algorithm 1.

---

**Algorithm 1:** FL at round $t$

---

1 **users side:**
2    **do in parallel for** $u \in \{1, \ldots, U\}$
3      Update $\boldsymbol{w}_t$ into $\boldsymbol{w}_t^u$ via, e.g., several training iterations;
4      Send to server $\boldsymbol{h}_t^u = \boldsymbol{w}_t^u - \boldsymbol{w}_t$;
5 **server side:**
6    Update $\boldsymbol{w}_t$ via equation 1;
7    Distribute $\boldsymbol{w}_{t+1}$ to all local users;
8 **return** *Updated global model, $\boldsymbol{w}_{t+1}$;*

---

**Algorithm 2:** FL + unlearning at round $t$

---

1 **initialization:** $\boldsymbol{w}_0$, $t$;
2 **for** $j \in \{0, \ldots, t\}$ **do**
3    Set $\boldsymbol{w}_{j+1}$ via Algorithm 1;
4    **server side:**
5      Store $\{\boldsymbol{h}_j^u\}_{u=1}^U$;
6 **server side:**
7    **if** *unlearning for user $\tilde{u}$* **then**
8      Compute $\boldsymbol{w}_{t+1}'$ via equation 3;
9      **return** *Unlearned model $\boldsymbol{w}_{t+1}'$;*
10 **return** *Updated global model, $\boldsymbol{w}_{t+1}$;*

---

**Distributed Unlearning.** Distributed unlearning extends the distributed learning framework to ensure the RTBF of its users upon request, as well as the ability detach maliciously injected backdoors once revealed. The goal of unlearning here is to erase the contributions of a user (or a group of users) while preserving the performance of the model acquired using the remaining clients Romandini et al. (2024). To formulate this, consider a distributed learning procedure that iterated over $t > 1$ rounds up to the arrival of the unlearning request regarding the $\tilde{u}$th user, $1 \leq \tilde{u} \leq U$. The desired unlearned model, coined the *train-from-scratch* model, is the one obtained by naively retraining the global model using all users except for the omitted user $\tilde{u}$ Liu et al. (2023), i.e., by iterating over

$$\boldsymbol{w}_{t+1}^\star = \boldsymbol{w}_t^\star + \frac{1}{U-1} \sum_{u=1, u \neq \tilde{u}}^{U} \boldsymbol{h}_t^{\star,u}, \quad \boldsymbol{h}_t^{\star,u} := \boldsymbol{w}_t^{\star,u} - \boldsymbol{w}_t^\star; \quad \boldsymbol{w}_0^\star = \boldsymbol{w}_0. \tag{2}$$

As elaborated in section 2, retraining from scratch is often infeasible. Most existing FU works relax it by balancing between partially retraining the local models and the subtraction of the unlearned user past updates, as both $\boldsymbol{w}_{t+1}, \boldsymbol{w}_{t+1}^\star$ rely of accumulating model updates, according to equations 1 and 2, respectively. When focusing on unlearning carried out solely on the server-side, without users' retraining, a generic unlearning rule is based on the one proposed in Huynh et al. (2024):

$$\boldsymbol{w}_{t+1}' = \boldsymbol{w}_{t+1} + \sum_{j=0}^{t} (1+\alpha)^{t-j} \boldsymbol{\delta}_j', \quad \boldsymbol{\delta}_j' := \frac{1}{U} \left( \frac{1}{U-1} \sum_{u=1, u \neq \tilde{u}}^{U} \boldsymbol{h}_j^u - \boldsymbol{h}_j^{\tilde{u}} \right). \tag{3}$$

In equation 3, $\alpha$ is a pre-determined *skewness* parameter, and $\boldsymbol{\delta}_j'$ represents the local update skew induced by client $\tilde{u}$ in round $j$. The overall procedure, carrying out $t$ training rounds followed by server-side unlearning of user $\tilde{u}$, is outlined as Algorithm 2.

**Problem Definition.** To be able to unlearn via equation 3, the server must store all past local updates for all users during training Cao et al. (2023); i.e., storing $\{\boldsymbol{h}_j^u\}$ for $1 \leq u \leq U$ and $0 \leq j \leq t$ .

Assuming each of the $m$ model parameters is represented using $b$ bits (e.g., $b = 64$ for a standard 64-bit floating point), FU involves storing $U \cdot t$ sequences of $m \cdot b$ bits. This induces a substantial burden for highly parameterized models (large $m$) trained over many rounds (large $t$) with data from numerous users (large $U$).

To mitigate this overhead, we aim to develop a FU framework, allowing compact storage of $\{\boldsymbol{h}_j^u\}$ by the server while minimally affecting its ability to unlearn. To formulate this mathematically, we are interested in lossy compression mechanisms $Q : \mathbb{R}^m \mapsto \mathbb{R}^m$ operating with a pre-defined compression rate $R \leq b$ (i.e., any entry of $Q(\boldsymbol{h}_j^u)$ is stored using $R$ bits), such that the model obtained via server-side unlearning with the compressed model updates $\{Q(\boldsymbol{h}_j^u)\}$, matches the train-from-scratch model. Since the server is unlikely to have prior knowledge of the model parameters distribution, we are interested in methods which are universal. Such schemes can be formulated as mappings of the model updates $\boldsymbol{h}_j^u$ into $Q(\boldsymbol{h}_j^u)$ stored by the server, while meeting requirements

> *R1* The lossy compression function is identical for all users and along time. This requirement significantly simplifies FU implementation.
>
> *R2* The scheme $Q(\cdot)$ must be invariant to the distribution of $\boldsymbol{h}_j^u$.

## 4 COMPRESSED DISTRIBUTED UNLEARNING

In this section we present candidate lossy compression techniques for achieving distributed unlearning with limited server-side memory footprint. We commence by formulating suitable lossy source coding approaches, after which we theoretically analyze their deviation from the optimal model.

### 4.1 LOSSY SOURCE CODING TECHNIQUES

Evidently, requirements *R1-R2* can be satisfied by any lossy source code that is invariant to the distribution of the model updates. A lossy source code is formulated as Polyanskiy & Wu (2014):

**Definition 4.1** (Lossy Source Code). *A lossy source code $Q(\cdot)$ with compression rate $R$, input size $L$, input alphabet $\mathcal{X}^L$, and output alphabet $\hat{\mathcal{X}}^L$, consists of:*

> *1. An encoder $e$: $\mathcal{X}^L \mapsto \{0, \ldots, 2^{LR} - 1\} := \mathcal{I}$ which maps the input into a discrete index.*
>
> *2. A decoder $d$: $\mathcal{I} \mapsto \hat{\mathcal{X}}^L$ which maps each $i \in \mathcal{I}$ into a codeword $\boldsymbol{q}_i \in \hat{\mathcal{X}}^L$.*

*For an input $\boldsymbol{x} \in \mathcal{X}^L$, the output of the code, $\hat{\boldsymbol{x}} \in \hat{\mathcal{X}}^L$, is written as $Q(\boldsymbol{x}) = d(e(\boldsymbol{x})) = \hat{\boldsymbol{x}}$.*

The performance of a lossy source code is characterized using its rate $R$ and distortion, the latter commonly being the mean-squared error (MSE), i.e., $\frac{1}{L}\mathbb{E}\big[\|\boldsymbol{x} - \hat{\boldsymbol{x}}\|^2\big]$. The encoder and decoder mappings are used to formulate the compressed FU rule, such that equation 3 becomes

$$\boldsymbol{w}_{t+1}'' = \boldsymbol{w}_{t+1} + \sum_{j=0}^{t}(1+\alpha)^{t-j}\boldsymbol{\delta}_j'', \quad \boldsymbol{\delta}_j'' := \frac{1}{U}\left(\frac{1}{U-1}\sum_{u=1,u\neq\tilde{u}}^{U}Q(\boldsymbol{h}_j^u) - Q(\boldsymbol{h}_j^{\tilde{u}})\right). \quad (4)$$

In accordance, FU in Algorithm 2 is reformulated into compressed FU in Algorithm 3.

While definition 4.1 is relatively general, we next focus on two family of lossy source codes, previously considered for FL in the context of communication efficiency Chen et al. (2021). These include codes that $(i)$ limit the volume of model updates by quantization, i.e., discretizing the updates such that they are expressed using a small number of bits Bernstein et al. (2018); Wen et al. (2017); Alistarh et al. (2017); Reisizadeh et al. (2020); and $(ii)$ save only part of the model updates by sparsification Konečný et al. (2016); Lin et al. (2017); Hardy et al. (2017); Aji & Heafield (2017); Alistarh et al. (2018); Han et al. (2020). The considered mechanisms are summarized in Table 1.

### 4.1.1 QUANTIZATION

Any lossy source code can be viewed as quantization Gray & Neuhoff (1998). However, the term *quantizers* typically refers to lossy source codes that operate block-wise, dividing an $m$-dimensional

vector $\boldsymbol{h}$ into $\lceil \frac{m}{L} \rceil$ blocks of size $L$, and encoding each block with the same code. When $L = 1$, the lossy source code implements *scalar quantization*, and $L > 1$ is termed *vector quantization*. While quantizers can be optimized to achieve improved rate-distortion tradeoff by tuning the code based on the input distribution, we are particularly interested in quantizers that are invariant of such distribution, and are therefore *universal*, meeting *R2*.

**Lattice Quantization.** A generic approach for universally choosing $L$-dimensional codewords is to realize them as the points of a *lattice* Zamir (2014). A truncated lattice is a set of points $\mathcal{P} := \{\boldsymbol{G}\boldsymbol{z} : \boldsymbol{z} \in \mathbb{Z}^L, \|\boldsymbol{G}\boldsymbol{z}\| < \gamma\}$, where $\boldsymbol{G}$ is an $L \times L$ non-singular matrix and $\gamma$ is a truncated sphere radios. A lattice quantizer $Q_{\mathcal{P}}(\cdot)$ maps each $\boldsymbol{x} \in \mathbb{R}^L$ into its nearest lattice point, i.e.,

$$Q_{\mathcal{P}}(\boldsymbol{x}) = \arg\min_{\boldsymbol{l} \in \mathcal{P}} \|\boldsymbol{x} - \boldsymbol{l}\|. \tag{5}$$

In general, the basic cell shape of a lattice can take different forms, such as hexagons for two-dimensional hexagonal lattices. When $\boldsymbol{G} = \Delta \cdot \boldsymbol{I}_{L \times L}$ for some $\Delta > 0$, $Q_{\mathcal{P}}(\cdot)$ realizes a scalar uniform quantizer per entry. The rate of $Q_{\mathcal{P}}(\cdot)$ is $R = \frac{1}{L} \log_2(|\mathcal{P}|)$, for $|\mathcal{P}|$ being the cardinality of $\mathcal{P}$. The overall number of bits required for storing an update vector $\boldsymbol{h} \in \mathbb{R}^m$ is thus $m \cdot R$ bits.

**(Probabilistic) Dithered Lattice Quantization.** The quantized representation in equation 5 and its error $\boldsymbol{e} := Q(\boldsymbol{x}) - \boldsymbol{x}$ are deterministically determined by the input $\boldsymbol{x}$. Nonetheless, by leveraging a random dither vector $\mathbf{d}$, $Q_{\mathcal{P}}(\cdot)$ can be extended into a probabilistic form via dithered quantization (DQ), yielding $Q_{\mathcal{P}}^{\mathrm{DQ}}(\cdot)$, and subtractive DQ (SDQ), yielding $Q_{\mathcal{P}}^{\mathrm{SDQ}}(\cdot)$, respectively; defined as

$$Q_{\mathcal{P}}^{\mathrm{DQ}}(\boldsymbol{x}) := Q_{\mathcal{P}}(\boldsymbol{x} + \mathbf{d}), \qquad (6a) \qquad Q_{\mathcal{P}}^{\mathrm{SDQ}}(\boldsymbol{x}) := Q_{\mathcal{P}}(\boldsymbol{x} + \mathbf{d}) - \mathbf{d}. \qquad (6b)$$

Proper selection of $\mathbf{d}$ can transform the quantization error $\boldsymbol{e}$ into a form of noise that is uncorrelated with the input Gray & Stockham (1993). One such setting is when $\mathbf{d}$ is uniformly distributed over the basic lattice cell, which is the set of points in $\mathbb{R}^L$ that are closer to $\boldsymbol{0}$ than to any other lattice point, defined as $\mathcal{P}_0 := \{\boldsymbol{x} : \|\boldsymbol{x}\| < \|\boldsymbol{x} - \boldsymbol{p}\| \; \forall \boldsymbol{p} \in \mathcal{P} \backslash \{\boldsymbol{0}\}\}$. Then, the quantization error is made independent of the quantized values, as stated in the following theorem Zamir & Feder (1996):

**Theorem 4.1.** *When $\mathbf{d}$ is uniformly distributed over $\mathcal{P}_0$ and $\boldsymbol{x}$ lies within the lattice support, i.e.,* $\Pr(\|\boldsymbol{x}\| \leq \gamma) = 1$, *then $\mathbf{e} = Q_{\mathcal{P}}^{\mathrm{SDQ}}(\boldsymbol{x}) - \boldsymbol{x}$ is uniformly distributed over $\mathcal{P}_0$ and independent of $\boldsymbol{x}$.*

When Theorem 4.1 holds, the error is clearly unbiased, as $\mathbb{E}[\mathbf{e}|\boldsymbol{x}] = 0$, and has a bounded variance, as $\mathbb{E}[\|\mathbf{e}\|^2|\boldsymbol{x}] = \mathbb{E}[\|\boldsymbol{d}\|^2]$. A similar result with higher error can be obtained with DQ, as stated in the following theorem Kirac & Vaidyanathan (1996):

**Theorem 4.2.** *When $\mathbf{d}$ is the sum of two mutually independent random vectors, each uniformly distributed over $\mathcal{P}_0$, and $\boldsymbol{x}$ lies within the lattice support, i.e.,* $\Pr(\|\boldsymbol{x}\| \leq \gamma) = 1$, *then $\mathbf{e} = Q_{\mathcal{P}}^{\mathrm{DQ}}(\boldsymbol{x}) - \boldsymbol{x}$ satisfies $\mathbb{E}[\mathbf{e}|\boldsymbol{x}] = 0$ and $\mathbb{E}[\|\mathbf{e}\|^2|\boldsymbol{x}] = \frac{3}{2}\mathbb{E}[\|\boldsymbol{d}\|^2]$.*

In compressed server-side unlearning, all forms of lattice quantization use $\log_2(|\mathcal{P}|)$ for storing every $L$-sized sub-vector of each model update. Nevertheless, SDQ also requires the server to store the dither signal $\boldsymbol{d}$, as it is used in decoding. This requirement can be alleviated via pseudo-random methods, obtaining random quantities realizations not by storing them directly, but rather by storing a single seed Shlezinger et al. (2020), such that its excessive storage is made negligible.

### 4.1.2 SPARSIFICATION

Sparsification is a form of lossy compression that discards a subset of its input. To formulate its operation, we focus on two $k$-sparse vectors operators, stated in Stich et al. (2018):

**Definition 4.2** (Sparsifier). *For a given $k \in [m]$, denote by $\Omega_k = \binom{[m]}{k}$ the set of all $k$-element subsets of $[m]$. Define $\omega \overset{\mathrm{u.a.r}}{\sim} \Omega_k$, and $\pi$ to be a permutation of $[m]$, satisfying $(|\boldsymbol{x}|)_{\pi(j)} \geq (|\boldsymbol{x}|)_{\pi(j+1)}$, for $j \in [m-1]$. Then, for $\boldsymbol{x} \in \mathbb{R}^m$, the mapping $\mathrm{spar}_k(\cdot) : \mathbb{R}^m \mapsto \mathbb{R}^m$, such that $\mathrm{spar} \in \{\mathrm{top}, \mathrm{rand}\}$, is given for each $i \in [m]$ by,*

$$(\mathrm{top}_k(\boldsymbol{x}))_i := \begin{cases} (\boldsymbol{x})_{\pi(i)}, & \textit{if } i \leq k, \\ 0, & \textit{otherwise,} \end{cases} \quad (7a) \qquad (\mathrm{rand}_k(\boldsymbol{x}))_i := \begin{cases} (\boldsymbol{x})_i, & \textit{if } i \in \omega, \\ 0, & \textit{otherwise,} \end{cases} \quad (7b)$$

**Algorithm 3:** FL + compressed unlearning

1 **initialization:** $\boldsymbol{w}_0$, $t$;
2 **for** $j \in \{0, \ldots, t\}$ **do**
3     Set $\boldsymbol{w}_{j+1}$ via Algorithm 1;
4     **server side:**
5        Store $\{e\left(\boldsymbol{h}_j^u\right)\}_{u=1}^U$;

6 **server side:**
7     **if** *unlearning for user $\tilde{u}$* **then**
8        Decode all updates $\{d\left(e\left(\boldsymbol{h}_j^u\right)\right)\}$;
9        Compute $\boldsymbol{w}_{t+1}''$ via equation 4;
10        **return** *Unlearned model $\boldsymbol{w}_{t+1}''$*;

11 **return** *Updated global model, $\boldsymbol{w}_{t+1}$*;

| Method | Def. | Required memory for each $\boldsymbol{h}_t^u \in \mathbb{R}^m$ | Stored seed |
|---|---|---|---|
| Lattice Q. | 5 | | ✗ |
| DQ | 6a | $\lceil \frac{m}{L} \rceil \log_2(|\mathcal{P}|)$ | ✗ |
| SDQ | 6b | | ✓ |
| $\text{top}_k$ | 7a | $k \cdot b + m$ | ✗ (✓*) |
| $\text{rand}_k$ | 7b | $k \cdot b$ | ✓ |

Table 1: Covered lossy source coding techniques. ✓* denotes that a seed is stored if random projections are used.

By Definition 4.2, for any $\boldsymbol{x} \in \mathbb{R}^m$, $\text{top}_k(\boldsymbol{x}) \in \mathbb{R}^m$ selects the top $k$ largest elements of $\boldsymbol{x}$ (in terms of their absolute value) with corresponding indices, while $\text{rand}_k(\boldsymbol{x}) \in \mathbb{R}^m$ uniformly at random selects $k$ elements from $\boldsymbol{x}$. In both cases, the remaining $m - k$ elements are set to zero Shi et al. (2019). $k$ is often defined as a rounded percent of $m$, i.e., $k = \lceil \zeta \cdot m \rceil$ for $0 < \zeta < 1$. In order to encode $\text{top}_k(\boldsymbol{x})$, the server stores the $k$ non-zero elements in full-precision. In decoding, an additional bit per entry is needed to identify which elements are set to zero. Therefore, assuming $b$-bit floating point representation, encoding requires $R = \frac{k}{m} \cdot b + 1$ bits per sample. For $\text{rand}_k(\cdot)$, where the sparsified indices do not depend on the input, one can potentially re-realize the random pattern $\omega$ in decoding using a pre-stored seed, similarly to SDQ, hence results in rate $R = \frac{k}{m} \cdot b$.

**Sparsification via random projections.** Sparsifiers are often employed in combination with random projections Konečný et al. (2016). Here, the input is first randomly projected using a stochastic unitary matrix before being sparsified. Decoding then consists of reconstructing the sparse vector and consequently multipling by the inverse projection. Formally, for a random unitary matrix $\mathbf{U}$ and a sparsifier $\text{spar}_k(\cdot)$ in Definition 4.2, equation 7 changes into $\text{spar}_k(\boldsymbol{x}) \mapsto \mathbf{U}^T \text{spar}_k(\mathbf{U}\boldsymbol{x})$. It is motivated as means to recover the input's most information before eliminating most of its entries. Unitary projection does not induce further computational overhead in calculating its inverse. While decoding requires access to $\mathbf{U}$, the fact that it is random and does not depend on the input indicates that it can be reconstructed by storing a seed rather than a full matrix for each model update.

## 4.2 THEORETICAL EVALUATION[1]

The induced distortion of lossy compression is inevitably incorporated into the unlearned model. To characterize this effect on the unlearned model performance in terms of forgetting capabilities, we use the conventional metric in unlearning literature Wang et al. (2024); Wu et al. (2020); Cao et al. (2023) of $L_2$-norm proximity to the train-from-scratch model, namely, $\|\boldsymbol{w}_{t+1}^\star - \boldsymbol{w}_{t+1}''\|$.

**Assumptions.** In our analysis we further adopt the following assumptions for $\forall t \geq 0, u \in [U]$:

*AS1*   Local training follows gradient decent, i.e., $\boldsymbol{w}_t^u := \boldsymbol{w}_t - \eta_t \nabla \mathcal{L}_u(\boldsymbol{w}_t)$; where $\eta_t > 0$ is the learning rate. Unlearning is realized using equation 4 with skewness $\alpha = 0$.

*AS2*   The norm of gradients is uniformly bounded, i.e., $\|\nabla \mathcal{L}_u(\boldsymbol{w}_t)\| \leq M$.

*AS3*   The distortions induced by the lossy source code, $\{\mathbf{e}_t^u := Q(\boldsymbol{h}_t^u) - \boldsymbol{h}_t^u\}$, are independent in time ($t$) and between users ($u$), and hold $\mathbb{E}[\mathbf{e}_t^u | \boldsymbol{h}_t^u] = 0$; $\mathbb{E}[\|\mathbf{e}_t^u\|^2 | \boldsymbol{h}_t^u] = \sigma^2$.

*AS1* focuses our analysis on a basic form of distributed learning, utilizing full gradient decent, and specializing a generic server-side unlearning obtained by subtracting the unlearned user past gradients. *AS2* is commonly adopted in distributed learning convergence studies Li et al. (2019); Stich (2018); Koloskova et al. (2019), and hold for, e.g., $L_2$-norm regularized linear regression and logistic regression objective functions. *AS3* is satisfied by different forms of probabilistic lossy compression,

---

[1]Proofs are deferred to the Appendix and appear in section A.1.

such as SDQ (Theorem 4.1) and DQ (Theorem 4.2). It is assumed here as the ability to represent distortion as additive noise notably facilitates analyzing its impact on the unlearning procedure, where aggregation in equation 4 results in this additive noise term effectively approaching its mean value of zero by the law of large numbers. Still, in our experimental study reported in section 5 we consider a broader range of lossy compression methods, including ones not necessarily holding *AS3*.

**Analysis.** The stochatsicty of the distortion (*AS3*) implies that the compressed unlearned model $\boldsymbol{w}_t''$ is a random vector. Thus, we fist formulate the first- and second-moments of the compression error:

**Lemma 4.1.** *Assuming* AS1 *and* AS3 *hold, given* $\{\mathcal{L}_u(\boldsymbol{w}_t)\}_{u,t}$, $\boldsymbol{w}_{t+1}''$ *in equation 4 is an unbiased estimator of the non-compressed* $\boldsymbol{w}_{t+1}'$ *in equation 3, i.e.,* $\mathbb{E}\left[\boldsymbol{w}_{t+1}'' - \boldsymbol{w}_{t+1}'\right] = 0$*; with variance*

$$\mathbb{E}\left[\left\|\boldsymbol{w}_{t+1}'' - \boldsymbol{w}_{t+1}'\right\|^2\right] = \frac{\sigma^2}{U-1}\sum_{j=0}^{t}\eta_j^2. \tag{8}$$

Using Lemma 4.1, we next derive an upper bound on the expected deviation from the desired model:

**Theorem 4.3.** *If* AS1-AS3 *hold, then given* $\{\mathcal{L}_u(\boldsymbol{w}_t)\}_{u,t}$*, the expected proximity of the train-from-scratch model* $\boldsymbol{w}_{t+1}^\star$ *to the compressed unlearned one* $\boldsymbol{w}_{t+1}''$ *in equations 2 and 4, respectively, obeys*

$$\mathbb{E}\left[\left\|\boldsymbol{w}_{t+1}^\star - \boldsymbol{w}_{t+1}''\right\|^2\right] \le G^2(t) + \frac{\sigma^2}{U-1}\sum_{j=0}^{t}\eta_j^2, \quad G(t) := 2M\sum_{j=0}^{t}\eta_j. \tag{9}$$

Theorem 4.3 characterizes the distortion induced by compressing the gradients used to form the unlearned model. This distortion, though, does not explicitly depend on the number of used bits, which is encapsulated in the second order moment of the used source code. It is further implied that Algorithm 3 does not guarantee decaying distance from the desired train-from-scratch model as the number of global iterations $t$ grow. This result is not surprising, as raised from the construction of the unlearned model in either equation 3 or equation 4, noticing that the remaining users past gradients $\{\mathcal{L}_u(\boldsymbol{w}_t)\}_{u\ne\tilde{u},t}$ still encapsulate information about the $\tilde{u}$th user via $\{\boldsymbol{w}_t\}_t$ in equation 1. However, the deviation can be made to converge as $t \to \infty$. Specifically, when the learning rate $\eta_t$ is carefully chosen to gradually decay over time, as highly adopted in other studies of distributed optimization Li et al. (2019); Stich (2018); Koloskova et al. (2019), we obtain the following theorem:

**Theorem 4.4.** *When the learning rate is set as* $\eta_t = \frac{1}{(t+\nu)^\lambda}$ *for* $t \ge 0$*;* $\nu > 0$*; and* $\lambda > 1$*, then there exists* $\bar{G} < \infty$ *such that* $\lim_{t\to\infty} G^2(t) = \bar{G}$ *and* $\lim_{t\to\infty}\mathbb{E}\left[\left\|\boldsymbol{w}_{t+1}^\star - \boldsymbol{w}_{t+1}''\right\|^2\right] \le \bar{G} + \frac{\sigma^2}{U-1}\left(\frac{1}{\nu} + \frac{\pi^2}{6}\right)$.

**Non-Compressed Analysis.** While the above results are derived for distributed unlearning with lossy compression, it also specializes non-compressed distributed unlearning as a special case. Specifically, by setting $Q(\boldsymbol{h}) \equiv \boldsymbol{h}$, we have $\boldsymbol{w}_{t+1}''$ realizes its non-compressed counterpart $\boldsymbol{w}_{t+1}'$, while *AS3* is satisfied as the distortion is the all-zero vector with probability one. In this case, we obtain a distinct bound on server-side unlearning for distributed gradient descent based learning:

**Corollary 4.1.** *When* AS1-AS2 *hold, the* $L_2$*-norm proximity of the unlearned model* $\boldsymbol{w}_{t+1}'$ *in equation 3 to the train-from-scratch one* $\boldsymbol{w}_{t+1}^\star$ *in equation 2 is given by*

$$\left\|\boldsymbol{w}_{t+1}^\star - \boldsymbol{w}_{t+1}'\right\| \le G(t), \tag{10}$$

*where* $G(t)$ *is given in equation 9 and is finite for* $t \to \infty$ *if* $\eta_t = \frac{1}{(t+\nu)^\lambda}$ *for* $t \ge 0$*;* $\nu > 0$*;* $\lambda > 1$*.*

Corollary 4.1 shows that, when combined with gradient descent for proper step-size setting, one can obtain convergent deviation from the desired model, as opposed to a generic divergent bound in Huynh et al. (2024). Contrasting it with Theorem 4.3 indicates that while the memory footprint of server side unlearning can significantly be relieved by the incorporation of lossy source coding, its excessive deviation decays not only with $\sigma$ (i.e., the quantization resolution), but also with the number of users $U$. This follows since, compared with Theorem 4.4, when $t \to \infty$, the excessive deviation due to compression is quantified to be not larger than $\frac{\sigma^2}{U-1}\left(\frac{1}{\nu} + \frac{\pi^2}{6}\right)$.

## 5 NUMERICAL EXPERIMENTS[2]

We next experimentally evaluate server-side unlearning with different forms of lossy compression. We utilize the unlearning procedure based on equation 3 with skewness parameter $\alpha = 0.07$ Huynh et al. (2024), comparing non-compressed unlearning to lossy compressed unlearning.

**Setting.** We evaluate compressed FU using two image classification datasets of MNIST and CIFAR-10. For each, we train a CNN composed of two convolutional layers and two fully-connected ones; with intermediate ReLU activations, max-pooling and normalization layers. The data distributed across $U = 25$ users in both *IID* and *non-IID* scenarios. The former equally partitions the data among clients, while the latter simulates label imbalance via the widely used Dirichlet distribution $\mathrm{Dir}(\beta)$ Li et al. (2022), having $\beta$ flexibly determines the imbalance level (smaller value leads to a higher unbalancedness). FL is globally iterated over 90 rounds, each with 10 randomly chosen clients locally utilizing SGD for 10 epochs.

**Unlearning Evaluation.** Unlearning request arrives at $t = 90$, due to the discovery of a participated malicious user that realized a backdoor attack during training. We employ the established edge-case backdoor Wang et al. (2020), where an adversary edge device intentionally uses wrong labels for a specific set of data points to mislead the server on seemingly easy inputs that are, though, unlikely to be part of the training (or test) data. This is illustrated in Fig. 1, where a digit $'7'$ is labeled as $'1'$ for MNIST; and an airplane is labeled as 'truck' for CIFAR-10. The unlearned model is then constructed at the server according to equation 4. Its performance is quantified via $(i)$ accuracy on the *primary task* test-set; and $(ii)$ accuracy on the *backdoor task* test-set, comprised of samples distribute similarly to those used by the attacker in training. Both preferably being as high, and low, respectively, as possible, with the latter being high indicates that the unlearned model is still backdoored Wang et al. (2024).

**Baselines.** We comparatively assess the performance of nine schemes, encompassing the lossy source codes covered in section 4.1. The three reference baselines are *vanilla FL*, i.e., the converged model $\boldsymbol{w}_{t+1}$ of the FL training in equation 1; *retrain*, denoting the desired train-from-scratch model $\boldsymbol{w}_{t+1}^{\star}$ in equation 2; and *non-compressed FU*, which uses the unlearned model $\boldsymbol{w}_{t+1}'$ obtained via equation 3. These are compared to the compressed unlearned model $\boldsymbol{w}_{t+1}''$ obtained via equation 4 using $Q_{\mathcal{P}}, L = l$; $Q_{\mathcal{P}}^{\mathrm{SDQ}}, L = l$; $\mathrm{top}_k$; and $\mathrm{rand}_k$. Each is respectively realizing $Q = Q_{\mathcal{P}}$ in equation 5; $Q = Q_{\mathcal{P}}^{\mathrm{SDQ}}$ in equation 6b; $Q = \mathrm{top}_k$ in equation 7a; and $Q = \mathrm{rand}_k$ in equation 7b. For the quantizers, $l \in \{1, 2\}$ and the lattice $\mathcal{P}$ is the standard hexagonal lattice. For sparsification, the sparsity level $k$ achieves the same rate as that of quantization based on Table 11.

**Results.** We begin with the MNIST IID scenario, and inspect the excess distortion in the recovered model induced by integrating lossy source codes into distributed server-side unlearning, compared to non-compressed unlearning; versus the compression rate. To that aim, for a given rate, Fig. 2 depicts the averaged model signal-to-noise ratio (SNR) for non-compressed and compressed unlearning, given by $\mathrm{SNR} = 10 \log_{10} \left( \mathrm{Var}(\boldsymbol{w}_t') / \mathrm{Var}(\boldsymbol{w}_t' - \boldsymbol{w}_t'') \right)$ [dB] (higher is better). Fig. 2 evidences, as also discussed in section 4.2, that the random nature of SDQ is a contributing factor, so as leveraging multivariate over scalar compression, having $Q_{\mathcal{P}}^{\mathrm{SDQ}}, L = 2$ performs the best of all quantizers. Predictably, in all baselines the SNR increases with rate, and saturates for the quantizers. Such a situation is attributed with an overloaded quantizer, implying that its error is dominated by the inputs residing outside of its dynamic range (support), thus further increment of its resolution (rate) does not improve its performance from some point on. Whereas $\mathrm{rand}_k$ appears to be the worst-performer, $\mathrm{top}_k$ is the best one for relatively high compression rates, as it is invariant to the input's support.

Next, Fig. 3 illustrates the evaluation of all baselines on the backdoor task test-set (lower is better), for different compression rates. As expected, vanilla FL and retrain obtain the worst and best results, being entirely affected and non-affected by the attacker participated in training, respectively. Non-compressed FU is shown to be the second-best, accurately discarding the malicious user influence. The compression counterparts show that quantizers yield improved unlearning over sparsifiers in all examined rates. In particular, the accuracy of non-compressed FU is degraded by about $2\%$ for $R \approx 4$, where for each user and FL iteration, any of the $m$ model updates is stored using $b$ bits for

---

[2]Additional numerical studies appear in section A.2.

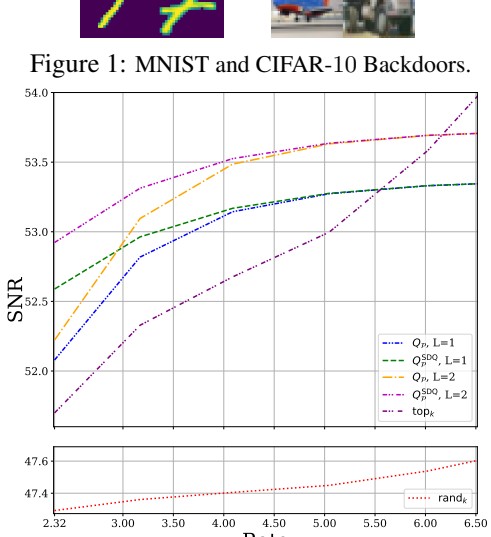

Figure 1: MNIST and CIFAR-10 Backdoors.

Figure 2: SNR ($\uparrow$) [dB] vs. rate [bits/sample].

Figure 3: Backdoor test-set accuracy ($\downarrow$)[%] vs. rate [bits/sample].

the former and $\lceil \frac{m}{2} \rceil \times 4$ for the latter. As a result, considering, e.g., a standard double-precision with $b = 64$, non-compressed FU memory footprint is $32\times$ higher than one that realizes $Q_{\mathcal{P}}^{\text{SDQ}}, L = 2$.

Finally, Table 2 summarizes the performance of all baselines for both IID and non-IID cases in each MNIST and CIFAR-10 datasets, respectively experimented with $\beta = 2, 1$ and rate $R = 2.8, 2.3$. Table 2 provides complementary evaluation to Fig. 3, as it also reports the accuracy on the primary task (higher is better). The inferior performance of the quantizers compared to the non-compressed alternative in the backdoor task visualized in Fig. 3 is here translated into an improved accuracy on the main task; yet less distinctive to a particular lossy source code to perform best. This, in turn, gives rise to the existence of a trade-off between memory footprint, accuracy on the primary task, and backdoor resiliency once (compressed) unlearning is carried out using (equation 4) equation 3.

| | MNIST | | | | CIFAR-10 | | | |
| | IID | | non-IID | | IID | | non-IID | |
| Method | Main | Attack | Main | Attack | Main | Attack | Main | Attack |
|---|---|---|---|---|---|---|---|---|
| vanilla FL | 99.15 | 75.53 | 99.12 | 52.10 | 71.48 | 32.83 | 66.49 | 28.30 |
| non-compressed FU | 96.07 | 0.27 | 96.30 | 0.0 | 53.89 | 2.64 | 30.39 | 1.89 |
| $Q_{\mathcal{P}}, L = 1$ | 98.58 | 3.99 | 98.1 | 1.47 | 62.93 | 3.96 | 45.45 | 1.32 |
| $Q_{\mathcal{P}}^{\text{SDQ}}, L = 1$ | 98.57 | 3.72 | 98.0 | 0.63 | 62.82 | 3.77 | 44.77 | 1.13 |
| $Q_{\mathcal{P}}, L = 2$ | 98.52 | 3.72 | 98.09 | 1.05 | 62.6 | 3.58 | 45.19 | 1.32 |
| $Q_{\mathcal{P}}^{\text{SDQ}}, L = 2$ | 98.51 | 3.19 | 97.93 | 0.42 | 62.27 | 3.58 | 44.20 | 0.94 |
| $\text{top}_k$ | 98.26 | 23.40 | 98.22 | 12.61 | 64.50 | 17.36 | 41.41 | 28.11 |
| $\text{rand}_k$ | 98.18 | 70.48 | 99.13 | 47.69 | 71.59 | 32.07 | 66.45 | 26.60 |

Table 2: Main task ($\uparrow$) and backdoor attack task ($\downarrow$) test-set accuracy [%].

## 6 CONCLUSIONS

In this work, we studied decentralized server-side unlearning with lossy source coding incorporated in the process. We investigated its effect on the unlearned model performance from both theoretical and experimental perspectives. From an experimental point of view, our numerical evaluations reveal that compressed FU with notable storage reduction, e.g., $32\times$ lower memory footprint at the server, preserves the ability to unlearn. On the theoretical side; we prove that under common assumptions, the distance between the compressed unlearned model to the desired one is bounded.

Our bound improves upon the best known guarantees, asserting that the growth rate of this distance is at most exponential with the number of FL iterations. However, as shown in recent not-limited-to-server FU works, in some scenarios the decentralized unlearned model may actually converge to the optimal one, and therefore our theoretical analysis may be a first important step towards a proof of convergence for server-side unlearning. We leave it as a main open question for future study.

Beyond extending our theoretical analysis, many interesting research directions are left for future work. For example, while our analysis considers basic gradient-based learning, one can potentially extend its findings to other forms of learning algorithms. Other important aspects to study concern the joint usage of sparsification and quantization; as well as unifying lossy and lossless compression, where the latter is known to provide further performance benefits from the underlying characteristics of the digital representations. Finally, prospective direction would try to mitigate the overloaded quantizers numerically evidenced in section 5 by designing them Shlezinger et al. (2019) or their associated lattices Lang et al. (2024) in a learning-oriented rather than a distortion-oriented manner.

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

# A APPENDIX

## A.1 DEFERRED PROOFS

### A.1.1 PROOF OF LEMMA 4.1

To prove Lemma 4.1, first note that iterating over equation 1 results in

$$\boldsymbol{w}_{t+1} = \boldsymbol{w}_t + \frac{1}{U}\sum_{u=1}^{U}\boldsymbol{h}_t^u = \boldsymbol{w}_0 + \sum_{j=0}^{t}\frac{1}{U}\sum_{u=1}^{U}\boldsymbol{h}_j^u. \tag{A.1.1}$$

Now, observing equation 3,

$$\boldsymbol{w}'_{t+1} = \boldsymbol{w}_{t+1} + \sum_{j=0}^{t}(1+\alpha)^{T-1-j}\delta_j$$

$$\overset{(a)}{=} \boldsymbol{w}_0 + \sum_{j=0}^{t}\frac{1}{U}\sum_{u=1}^{U}\boldsymbol{h}_j^u + \sum_{j=0}^{t}(1+\alpha)^{T-1-j}\frac{1}{U}\left(\frac{1}{U-1}\sum_{\substack{u=1\\u\neq\tilde{u}}}^{U}\boldsymbol{h}_j^u - \boldsymbol{h}_j^{\tilde{u}}\right)$$

$$\overset{(b)}{=} \boldsymbol{w}_0 + \frac{1}{U-1}\sum_{j=0}^{t}\sum_{\substack{u=1\\u\neq\tilde{u}}}^{U}\boldsymbol{h}_j^u = \boldsymbol{w}_0 + \frac{1}{U-1}\sum_{j=0}^{t}\sum_{\substack{u=1\\u\neq\tilde{u}}}^{U}(\boldsymbol{w}_t^u - \boldsymbol{w}_t)$$

$$\overset{(c)}{=} \boldsymbol{w}_0 - \sum_{j=0}^{t}\eta_j\frac{1}{U-1}\sum_{\substack{u=1\\u\neq\tilde{u}}}^{U}\nabla\mathcal{L}_u(\boldsymbol{w}_j), \tag{A.1.2}$$

here $(a)$ follows by equation A.1.1 and the definition of $\boldsymbol{\delta}_j$ in equation 3; and $(b)$, $(c)$ are true from *AS1*, as $\alpha = 0$ and $\boldsymbol{w}_t^u := \boldsymbol{w}_t - \eta_t\nabla\mathcal{L}_u(\boldsymbol{w}_t)$, respectively. $\boldsymbol{w}''_{t+1}$ in equation 4 similarly changes, and

$$\boldsymbol{w}''_{t+1} - \boldsymbol{w}'_{t+1} = \boldsymbol{w}_0 - \sum_{j=0}^{t}\eta_j\frac{1}{U-1}\sum_{\substack{u=1\\u\neq\tilde{u}}}^{U}Q\left(\nabla\mathcal{L}_u(\boldsymbol{w}_j)\right) - \left(\boldsymbol{w}_0 - \sum_{j=0}^{t}\eta_j\frac{1}{U-1}\sum_{\substack{u=1\\u\neq\tilde{u}}}^{U}\nabla\mathcal{L}_u(\boldsymbol{w}_j)\right)$$

$$= \sum_{j=0}^{t}\eta_j\frac{1}{U-1}\sum_{\substack{u=1\\u\neq\tilde{u}}}^{U}\left(Q\left(\nabla\mathcal{L}_u(\boldsymbol{w}_j)\right) - \nabla\mathcal{L}_u(\boldsymbol{w}_j)\right) = \sum_{j=0}^{t}\eta_j\frac{1}{U-1}\sum_{\substack{u=1\\u\neq\tilde{u}}}^{U}\mathbf{e}_u^j.$$

According to Lemma 4.1, $\{\mathcal{L}_u(\boldsymbol{w}_t)\}_{u,t}$ are given, and therefore, due to *AS3*, $\forall t, u \; \mathbb{E}\left[\mathbf{e}_u^t\right] = 0$ and $\mathbb{E}\left[\|\mathbf{e}_u^t\|^2\right] = \sigma^2$, which implies that $\mathbb{E}\left[\boldsymbol{w}''_{t+1} - \boldsymbol{w}'_{t+1}\right] = 0$. As for the second moment, since *AS3* states that $\forall t, u \; \{\mathbf{e}_u^t\}$ are independent, we desirably obtain

$$\mathbb{E}\left[\left\|\boldsymbol{w}''_{t+1} - \boldsymbol{w}'_{t+1}\right\|^2\right] = \mathbb{E}\left[\left\|\sum_{j=0}^{t}\eta_j\frac{1}{U-1}\sum_{\substack{u=1\\u\neq\tilde{u}}}^{U}\mathbf{e}_u^j\right\|^2\right] = \sum_{j=0}^{t}\eta_j^2\frac{1}{(U-1)^2}\sum_{\substack{u=1\\u\neq\tilde{u}}}^{U}\mathbb{E}\left[\|\mathbf{e}_u^j\|^2\right]$$

$$= \frac{\sigma^2}{U-1}\sum_{j=0}^{t}\eta_j^2.$$

### A.1.2 Proof of Theorem 4.3

To prove Theorem 4.3, we characterize the intermediate distances to the non-compressed model $\boldsymbol{w}'_{t+1}$ in equation 3 and utilizing Lemma 4.1. Namely,

$$\mathbb{E}\left[\left\|\boldsymbol{w}_{t+1}^\star - \boldsymbol{w}''_{t+1}\right\|^2\right] = \mathbb{E}\left[\left\|\boldsymbol{w}_{t+1}^\star - \boldsymbol{w}'_{t+1}\right\|^2\right] + \mathbb{E}\left[\left\|\boldsymbol{w}'_{t+1} - \boldsymbol{w}''_{t+1}\right\|^2\right]$$

$$+ 2\left\langle \boldsymbol{w}_{t+1}^\star - \boldsymbol{w}'_{t+1}, \mathbb{E}\left[\boldsymbol{w}'_{t+1} - \boldsymbol{w}''_{t+1}\right]\right\rangle = \left\|\boldsymbol{w}_{t+1}^\star - \boldsymbol{w}'_{t+1}\right\|^2 + \frac{\sigma^2}{U-1}\sum_{j=0}^{t}\eta_j^2. \tag{A.1.3}$$

The proof is then concluded by proving the following auxiliary lemma:

**Lemma A.1.1.** *When* AS1 *and* AS2 *hold, the distance between the train-from-scratch model $\boldsymbol{w}_{t+1}^\star$ in equation 2 and the unlearned one $\boldsymbol{w}'_{t+1}$ in equation 3 satisfies*

$$\left\|\boldsymbol{w}_{t+1}^\star - \boldsymbol{w}'_{t+1}\right\| \le 2M\sum_{j=0}^{t}\eta_j.$$

*Proof.* Under *AS1*, we use the formulation of $\boldsymbol{w}'_{t+1}$ in equation A.1.2, according to which we obtain similar representation for $\boldsymbol{w}^\star_{t+1}$, thus

$$
\|\boldsymbol{w}^\star_{t+1} - \boldsymbol{w}'_{t+1}\| = \left\| \boldsymbol{w}_0 - \sum_{j=0}^{t} \eta_j \frac{1}{U-1} \sum_{\substack{u=1 \\ u \neq \tilde{u}}}^{U} \nabla \mathcal{L}_u(\boldsymbol{w}^\star_j) - \left( \boldsymbol{w}_0 - \sum_{j=0}^{t} \eta_j \frac{1}{U-1} \sum_{\substack{u=1 \\ u \neq \tilde{u}}}^{U} \nabla \mathcal{L}_u(\boldsymbol{w}_j) \right) \right\|
$$

$$
= \left\| \sum_{j=0}^{t} \eta_j \frac{1}{U-1} \sum_{\substack{u=1 \\ u \neq \tilde{u}}}^{U} \left( \nabla \mathcal{L}_u(\boldsymbol{w}^\star_j) - \nabla \mathcal{L}_u(\boldsymbol{w}_j) \right) \right\|
$$

$$
\overset{(a)}{\leq} \sum_{j=0}^{t} \eta_j \frac{1}{U-1} \sum_{\substack{u=1 \\ u \neq \tilde{u}}}^{U} \left( \|\nabla \mathcal{L}_u(\boldsymbol{w}^\star_j)\| + \|\nabla \mathcal{L}_u(\boldsymbol{w}_j)\| \right) \overset{(b)}{\leq} 2M \sum_{j=0}^{t} \eta_j,
$$

where $(a)$ holds by recursively applying the triangle inequality and $(b)$ is true from using *AS2* for every $u$ and $j$; concluding the proof. $\qquad\square$

### A.1.3  PROOF OF THEOREM 4.4

For $t \to \infty$, $G^2(t)$ converges if $\sum_{j=0}^{t} \eta_j$ does. Now, the setting of $\eta_t$ in Theorem 4.4 implies that

$$
\sum_{j=0}^{t} \eta_j = \sum_{j=0}^{t} \frac{1}{(j+\nu)^\lambda} \leq \frac{1}{\nu} + \sum_{j=1}^{t} \frac{1}{j^\lambda} \tag{A.1.4}
$$

where the inequality follows as $\nu > 0$, and additionally, the right-hand-side of equation A.1.4 is summable as $\lambda > 1$. To complete the proof we left to show that for $t \to \infty$ it holds that $\frac{\sigma^2}{U-1} \sum_{j=0}^{t} \eta_j^2 \leq \frac{\sigma^2}{U-1} \left( \frac{1}{\nu} + \frac{\pi^2}{6} \right)$. Using similar arguments,

$$
\sum_{j=0}^{\infty} \eta_j^2 = \sum_{j=0}^{\infty} \frac{1}{(j+\nu)^{2\lambda}} \leq \frac{1}{\nu} + \sum_{j=1}^{\infty} \frac{1}{j^2} = \frac{1}{\nu} + \frac{\pi^2}{6},
$$

as desired.

### A.2  ADDITIONAL EXPERIMENTS

This section presents additional experimental results on the MNIST dataset, highlighting the relationship between the compression rate and the resulted unlearned model ability to maintain high accuracy on the primary task; studied in the IID scenario. Furthermore, for the non-IID case, we examine the effect of the data imbalance level $\beta$ on the performance of the unlearned models in the backdoor task.

Fig. 4 complements Fig. 3 and Table 2 by illustrating the accuracy of all baselines on the primary task test-set (higher is better) under varying compression rates. As anticipated, retrain is the best baseline on the main task, completely not backdoored. Additionally, it is visualized that the unlearned model obtained for either of the compression methods, except for $\mathrm{rand_k}$, exhibits a decreased performance for growing rates. This follows as compressing with higher rate, i.e., an improved resolution, is attributed with lesser variance and therefore more reliable estimation of the input. In this case the input is the non-compresses FU model, which itself results in the lowest main task accuracy, in contract to being the best baseline on the backdoor task, as showed in Fig. 3 and Table 2.

Next, in Fig.5 we evaluate the performance of all baselines on the backdoor task test-set, with respect to varying values of $\beta$, which controls the degree of imbalance (lower values indicate greater imbalance), while keeping the compression rate $R = 2.3$ fixed. The top of the figure illustrates variations in the backdoor accuracy, which result from the fact that the number of backdoor samples is proportional to the attacker's local dataset size. Adjusting $\beta$ alters the number of samples allocated to each user, thereby influencing the backdoor attack performance on the vanilla FL model. As a result, we do not expect to observe consistent behaviour across different values of $\beta$ in terms of backdoor accuracy. However, we do observe persistent trends across all methods for a fixed $\beta$, keeping the best-to-worst performers order identical to that in Fig.3.

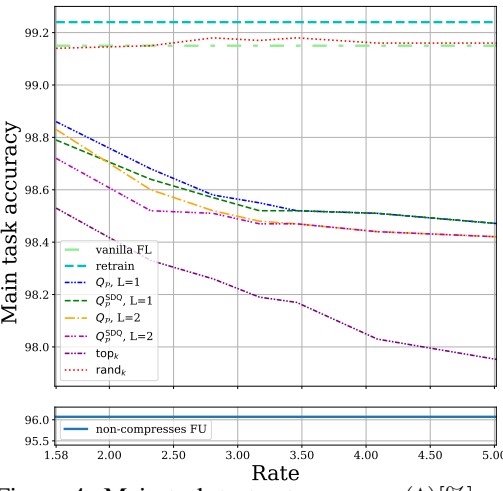

Figure 4: Main task test-set accuracy (↑)[%] vs. rate [bits/sample].

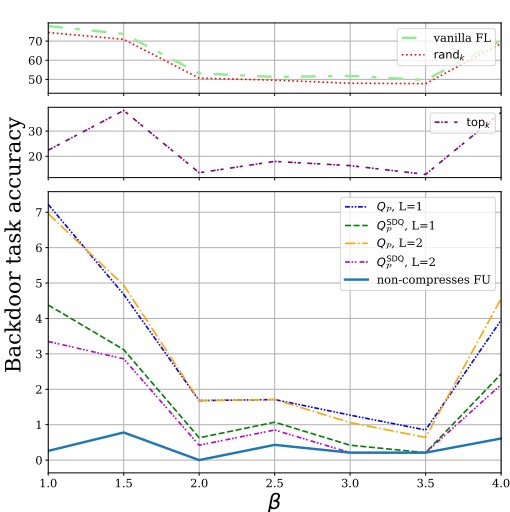

Figure 5: Backdoor test-set accuracy (↓)[%] vs. $\beta$ (imbalance level).

