# OpenReview forum: "Distributed Unlearning with Lossy Compression"
_ICLR.cc/2025/Conference — ICLR 2025 Conference Withdrawn Submission_

### Official Review · Reviewer_yQFk · 2024-10-27

**Soundness:** 3
**Presentation:** 3
**Contribution:** 2
**Rating:** 3
**Confidence:** 4

**Summary:**

The paper addresses lossy compression of update history in a distributed unlearning setup. The authors propose using lattice quantization and provide a theoretical bound on the differences between weights.

**Strengths:**

The paper is generally well-structured and accessible, aside from some issues with the citation style.

**Weaknesses:**

1. The reviewer is concerned about the paper’s limited novelty. Applying lattice quantization to the distributed unlearning problem appears to be a relatively straightforward adaptation of existing methods. Although the authors present a theoretical analysis (Theorem 4.3), it seems fairly direct. This is because the bound between the target weight $w$ and the compressed unlearned weight $w''$ can be expressed simply as $ \|w - w'\| + \|w' - w''\| $, where $ w'$  represents the uncompressed unlearned weight. This approach lacks significant innovation in its methodology since each term can be bounded by previous works.

2. Please revise the citation style to incorporate \citep or \citet where appropriate, and use ` ' for quotations.

3. For equation, it would be better to use \eqref

4. In equation 2 and 3, usage of $t$ seems confusing. My understanding is $t$ is for all iteration in equation 2, while $t$ represents the last iteration in equation 3.

5. Is there any analysis on sparsification?

**Questions:**

See Weaknesses.

---

> ### Author Response · Authors · 2024-11-20
>
> Thank you very much for the thoughtful review and insightful comments! We also appreciate the kind feedback regarding the structure and accessibility! Although the reviewer had no questions, we would like to discuss a few important points mentioned in his comments, which we aim to clarify and ensure accuracy. We shall address them in the same order in which they were indexed by the reviewer.
>
> **W1**: While this comment is rather subjective, let us try to convince the reviewer that our work is novel and of interest to the community, and that the techniques are not trivial, discussing each of which separately, as follows.
>
> **Relevance of the setup**. We tackle a relevant problem in distributed machine learning. There is a need for federated unlearning, which directly follows from the growing popularity of federated learning (what if some user is detected at some point, perhaps even much after training is concluded, as being malicious, compromised, buggy, or just wishes to be forgotten?). Storage here becomes a real bottleneck, making one question whether this can even be done (as raised by Reviewer F5mS).
>
> **Novelty of the solution**. The solution we propose – use lossy compression when storing – has not yet been considered for federated unlearning. It is indeed inspired by the employment of such mechanisms in other domains (including federated learning), but that is what makes it attractive and easy to apply.
>
> **The first to examine**. It is emphasized that we are, to the best of our knowledge, the first work that systematically examines the theoretical and numerical aspects of compressed distributed machine unlearning; in its most challenging setting of executing unlearning merely by the orchestrating server, which demands an excessive memory footprint. In previous papers considering decentralized unlearning (there are a number of such papers by now, see (Romandini et al 2024)), the prevalent framework is having the entity performing unlearning collaboratively being either of the server, target (unlearned) client, and remaining users; often involving some
> extent of training-from-scratch. This stands in a strike contrast to the fundament nature of federated learning, as user’s retraining or cooperation is not always feasible, due to it being no longer (or not willing to be) available or detected to be malicious, respectively.
>
> **Improving non-compressed unlearning**. One of the most surprising results in our paper is that our upper bound on the difference between the desired model that is trained from scratch and the model unlearned from lossy compressed stored updates outperforms the
> state-of-the-art known bounds for non-compressed decentralized server-side unlearning, even when lossy compression is incorporated. Our characterization also specializes a bound for non-compressed distributed unlearning. Before our work, it was not clear if such a bound can be even achieved. As an example, (Huynh et al 2024) only postulates that the growth rate of this difference (between non-compressed unlearning and train-from-scratch models) is at most exponential with the number of federated learning iterations. Thus, our results are much stronger. Finally, from an experimental point of view, the integration of compression is shown to provide robustness against different scenarios of backdoor attacks, compared to its more vulnerable non-compressed counterpart (please see also our detailed related response to Question 2 of Reviewer 3Dd1).
>
> **First step towards convergence**. In terms of convergence to the optimal model (rather than bounding the distance from it), recent not-limited-to-server server side federate unlearning works show that in some scenarios the decentralized unlearned model may actually converge to the optimal model. Therefore our theoretical analysis may be a first important step towards a proof of convergence for server-side unlearning.
>
> **Proof technique’s simplicity**. We are proud and excited about the fact that the proof technique is simple. We do not think that this is a weakness of our paper. Furthermore, we are pretty sure that our techniques can be applied to many interesting variants and generalization of out setup (e.g., the joint usage of sparsification and quantization, unifying lossy and lossless compression), and is thus expected to intrigue follow up works and further research.
>
> Following the reviewer’s comment, in the revised version of our paper, we will strike to further emphasize the novelty of our techniques and results compared to previous work.
>
> **W2**: Thanks! The citation style as well as the quotation marks will be fixed in the revised version.
>
> **W3**: Correct! In the revised version, we will use \eqref for equations not yet referenced using it.
>
> **W4**: Great point, thank you; you understood correctly. We will follow your suggestion and notate $t$ differently in equations 2 and 3.

---

> > ### Author Response · Authors · 2024-11-20
> >
> > **W5**: Great idea! We consider the sparsification analysis as a main direction for future work, and strongly believe that: (1) the unbiasedness and fixed variance of compression error in AS3 are not satisfied for the biased compressor top-k, that should be therefore analyzed under different regime of assumptions and (2) rand-k only satisfy these assumptions if the updates lie on a sphere (which is equivalent to lattice support for lattice quantization) (Zhao et al 2023). This adds up to the many possible future research directions arising from the novel framework of compressed federated unlearning. Following the reviewer’s comment, this would be discussed in the conclusions section of our revised paper.
> >
> > References
> > - (Romandini et al 2024) Federated unlearning: A survey on methods, design guidelines, and evaluation metrics. IEEE NNLS.
> > - (Huynh et al 2024) Fast-fedul: A training-free federated unlearning with provable skew resilience. ECML PKDD.
> > - (Zhao et al 2023) Towards efficient communications in federated learning: A contemporary survey. Journal of the Franklin Institute.

---

> ### Author Response · Authors · 2024-11-22
> **Thanks again for your review and time**
>
> Thanks again for your thoughtful review! We believe that we have addressed all of your concerns and questions in our response above. We would love to receive any additional feedback you may have. Do you have any follow-up questions? We are excited to engage in further discussions this week! Please let us know.
>
> Thank you very much, and we look forward to hearing from you.

---

> > ### Comment · Reviewer_yQFk · 2024-11-26
> >
> > Thank you for your detailed response. While I appreciate your clarifications and proposed revisions, my concerns about the limited novelty and incomplete analysis remain. I will maintain my original score.

---

### Official Review · Reviewer_3Dd1 · 2024-11-04

**Soundness:** 3
**Presentation:** 2
**Contribution:** 3
**Rating:** 6
**Confidence:** 2

**Summary:**

The paper addresses the problem of unlearning in a federated setting using compression techniques. Specifically, at a certain point during training, a client may wish to withdraw from the process and have its contributions removed. A brute-force solution would be to restart training from scratch without this client, but this is impractical. Another approach would be to save the information sent by each client at each iteration on the server so that, if a client opts out, the server can recompute the updated model using the saved information (in this case, locally model updates), leading to an approximate solution. However, saving model updates for each client at each iteration requires substantial memory, which becomes impractical with large models, many clients, or numerous iterations. To address this issue, the authors propose using different compression techniques that reduce the serve memory footprint. The paper provides theoretical guarantees for these approaches, with practical experiments on CIFAR-10 and MNIST to validate the theoretical findings.

**Strengths:**

- The paper is easy to read.
- Provides theoretical guarantees for using compression techniques in federated unlearning.
- Includes numerical experiments on MNIST and CIFAR-10 on IID and Non-IID data distribution demonstrating the method’s practicality, showing improved accuracies on the main task compared to non-compression unlearning, while still maintaining reasonable performance against backdoor attacks.

**Weaknesses:**

- Evaluation is conducted on a 2-layer model for CIFAR-10. Given that compression is used, wouldn’t it be more practical now to use a deeper CNN for unlearning?

- The practical evaluation focuses on backdoor attacks, which does not address the use case where a client simply wants to withdraw from training (without having performed backdoor attacks) and ensure that its information is no longer used. Have the authors considered an alternative metric to assess the effect of compression on the ability to detect whether a removed client contributed to the training?

**Questions:**

From Theorems 4.3 and 4.4, it appears that G(t) can be large, as it depends on the norm of the gradients. Do you have an idea of how large it tends to be in practice?

How practical would this approach be when using a model like ResNet-18 on CIFAR-10? What would the server's memory footprint be in that scenario?

From Table 2, it seems that compression results in better accuracy on the main task compared to non-compressed federated unlearning. Do you have an explanation or insights as to why this might be the case?

---

> ### Author Response · Authors · 2024-11-20
>
> Thank you very much for the thorough review and insightful comments! We also appreciate the kind feedback regarding presentation, theoretical guarantees, and practicality!
>
> **W1**: We thank the reviewer for this important comment. Indeed, while lowering the server footprint is advantageous in many various scenarios, e.g., enabling unlearning in limited storage capabilities parameter servers; it is of a crucial importance once highly parametrized models are concerned, such as deeper neural networks. Following the reviewer’s comment, in the revised version of our paper, we will further experiment deeper architectures to be learned, that more suitably reflect the necessity of compression for enabling unlearning; and add a short discussion.
>
> **W2**: Thank you for this constructive comment. Sever-side unlearning is undoubtedly introduced to remove the contributions of a client from a trained model, while preserving its usefulness. As such, it should be evaluated in terms of (i) the main task performance, i.e., the validity of the model; and (ii) the unlearning performance, i.e., that the contribution of the removed client was indeed canceled. While the former is simple to evaluate via, e.g., the test accuracy of the model on unseen data, assessing unlearning performance is more challenging (Romandini et al. 2024). One approach, which we also evaluate in the paper, is to compare performance to training from scratch, though similar performance may not necessarily indicate that the erased user was indeed forgotten. An alternative approach evaluates the model on the training data of the erased user, which again may not faithfully capture unlearning performance, as the
> model may perform accurately on such data even if its contribution was omitted from training, depending on the heterogeneity of the data. For this reason, we adopt the more challenging option, in which the user employs a backdoor attack, as it allows to faithfully evaluate both main task performance and unlearning performance, without being restricted to specific data distributions.
>
> **Q1**: Great point, thank you. Our experiments section reveals that the numerical value of $G(t)$ is practically implied to not be too large. That is, Figures 3-4 reflect the bound in the stated theorems by referencing the performance of compressed unlearning to the one obtained by the optimal train-from-scratch model; having both relatively close. Nevertheless, following the reviewer’s question, in the revised version of our paper, we empirically estimate the value of $G(t)$ for representative training processes.
>
> **Q2**: Please note that the server’s memory footprint for executing federated unlearning, for any considered model architecture, is unaffected by the used dataset; as it solely determined by the number of trainable weights. This is formally defined in the system model section of our manuscript to be $U \cdot t \cdot m \cdot b$, respectively denoting the number of: edge clients, global iterations, learnable parameters, and bits required to store each parameter. More specifically, considering the ResNet-$18$ model mentioned by the reviewer and a standard $64$-bit floating point, implies that $m = 11 \cdot 10^6$ and $b = 64$, respectively. Utilizing our approach via, e.g.,
> two dimensional lattice quantization with a bit rate of $4$ suggests that $b = \frac{4}{2} = 2$, and the footprint is accordingly $32\times$ lower. This highly beneficial footprint is also numerically validated in the experiments section of our submitted paper to not significancy deteriorate the model’s unlearning performance.
>
> **Q3**: Thank you for pointing this important note out. Indeed, the integration of compression in unlearning often results in better accuracy on the main task compared to non-compressed federated unlearning, as revealed in both Table 2 and Figure 4. To explain this, recall that according to Section 5, the experimental setting formulation assumes that a user employs a backdoor attack which only corrupts some specific labels. As a result, its contribution is actually also beneficial to the main task up to some extent. Consequently, its full cancellation via non-compressed federated unlearning is more harmful to our main task that canceling this user up to a distortion term, as the case in compressed federated unlearning. This, in turn, gives rise to the existence of a trade-off between memory footprint, accuracy on the primary task, and backdoor resiliency once (compressed) unlearning is carried out. Following the reviewer’s question, in the revised paper, we will add this explanation to the discussion accompanies Table 2.
>
> References
> - (Romandini et al 2024) Federated unlearning: A survey on methods, design guidelines, and evaluation metrics. IEEE NNLS.

---

> ### Author Response · Authors · 2024-11-22
> **Thanks again for your review and time**
>
> Thanks again for your thoughtful review! We believe that we have addressed all of your concerns and questions in our response above. We would love to receive any additional feedback you may have. Do you have any follow-up questions? We are excited to engage in further discussions this week! Please let us know.
>
> Thank you very much, and we look forward to hearing from you.

---

> > ### Comment · Reviewer_3Dd1 · 2024-11-26
> >
> > I thank the authors for their response and their willingness to add experiments on a deeper model in the next version of the paper.
> > I keep my score.

---

### Official Review · Reviewer_F5mS · 2024-11-09

**Soundness:** 3
**Presentation:** 2
**Contribution:** 2
**Rating:** 3
**Confidence:** 4

**Summary:**

The paper studies federated unlearning and adopts the method in [1]. As the approach in [1] (stated in (3) in this paper) requires the server to know all the historical updates from all the clients, it adds big storage cost. The paper proposes some compression techniques that guarantees unbiasedness and bounded error in the unlearning setup, in particular.


[1] Thanh Trung Huynh, Trong Bang Nguyen, Phi Le Nguyen, Thanh Tam Nguyen, Matthias Weidlich, Quoc Viet Hung Nguyen, and Karl Aberer. Fast-fedul: A training-free federated unlearning with provable skew resilience. In Joint European Conference on Machine Learning and Knowledge Discovery in Databases, pp. 55–72. Springer, 2024.

**Strengths:**

The theoretical derivations looked correct to me.

**Weaknesses:**

- While the proposed approach and theoretical derivations look okay to me when optimizing for the approach in (3), I am not convinced that (3) is a reasonable unlearning technique in federated learning. I don't think server storing all the historical updates is a practically feasible solution. Instead, if the clients directly send their data to the server (i.e. non-federated training), it could have yielded lower storage cost in many cases. Since the paper does not concern the federated learning systems with privacy mechanisms (which would make unlearning unnecessary), the main advantage of federated learning here is the decentralization. However, keeping all the updates at the server all the time kills this advantage. I know that this paper is trying to reduce this storage cost but I still can't imagine this unlearning approach (in (3)) having a practical use.

- The writing could be improved. A few formatting/grammar issues:

     - From the abstract "We further provide a numerical study, shows that..."
    - Please use inline references in parentheses when they are not part of the sentence. For example, the reference in "While data is often abundantly available in the ’big data’ era Jordan & Mitchell (2015), the source of the data might" is not part of the sentence and should be enclosed in parentheses.
    - Line 330: "fist" --> "first"

**Questions:**

If the authors have any justification for storing the updates at the server that I cannot think of, I am happy to discuss further and revisit my evaluation.

---

> ### Author Response · Authors · 2024-11-20
>
> Thank you very much for the thoughtful review and the kind feedback regarding theoretical derivations! We also appreciate the reviewer’s willingness to further discuss the justification for storing the updates at the server.
>
> **W1**: Following this comment, let us try to convince the reviewer at the necessity of the approach in (3) as an unlearning technique in federated learning; with, or without, privacy-preserving mechanisms integrated into the training process.
>
> **Unlearning use-case**. Please consider the following scenario, where a parametrized model has been trained in a federated manner with, or without, privacy-preserving mechanisms integrated into the process. Then, after the learning phase has concluded, the server discovered either of the following: (a) one of the participated clients has been revealed as malicious; (b) one the the users has requested to be forgotten; (c) the training process of some user had a bug. In each, a certain contribution needs to be undone. But, unfortunately, re-training the network is not an option, as the server may not be able to restart the federated learning procedure, and
> may not even have access to the users anymore.
>
> **Privacy cannot help here**. It should be emphasized that privacy-preserving federated learning does not address this issue. To see this, let us now narrow down the use-case described above to particularly focus on the federated training using privacy-preserving mechanisms, while one the client to be removed is only detected in the end of the learning process; having all contributed users already disconnected from the server. Then, in such a scenario, whether or not privacy-preserving mechanisms were present in training does not change (particularly if the client to be removed is malicious, and may not be playing by the rules).
>
> **Existing solution necessitate heavy storage**. Continuing with our running example, the server, with no connected devices, possesses a trained model from which it has to unlearn all past contributions of a certain user. While the optimal solution of training-from-scratch the
> model with all the rest of users is out of reach; the relaxed alternative the server could do is to somehow unlearn the erased client’s contributions from the trained model; if, of course, it has this past information saved. This requires the server to store all model updates observed in training, leading to immense storage overhead for preserving the ability to unlearn.
>
> **Compressed federated unlearning**. The above indicates (i) that server-side federated unlearning is a relevant problem, that directly follows from the growing popularity of federated learning; and (ii) that existing solutions may not be feasible, as noted by the reviewer. This issue is exactly what we aim to overcome in our paper. Our proposed solution, integrating established and simple-to-apply lossy compression mechanisms, is a first step towards making server-side federated unlearning, which is clearly desirable, a feasible and applicable approach.
>
> Following the reviewer’s comment, in the revised version of our paper, we will strike to further explain the motivation behind and desirability of our studied problem in the introduction section, and specifically exemplify it via the described use-case.
>
> **W2**: Thanks! All mentioned corrections will be fixed in the revised version. We have also carefully gone through the revised manuscript to make sure that it no longer contains any typographical, grammatical and citation-style errors.
>
> **Q1**: Please see our detailed related response to your first comment.

---

> ### Author Response · Authors · 2024-11-22
> **Thanks again for your review and time**
>
> Thanks again for your thoughtful review! We believe that we have addressed all of your concerns and questions in our response above. We would love to receive any additional feedback you may have. Do you have any follow-up questions? We are excited to engage in further discussions this week! Please let us know.
>
> Thank you very much, and we look forward to hearing from you.

---

### Official Review · Reviewer_Pw5D · 2024-11-12

**Soundness:** 3
**Presentation:** 2
**Contribution:** 1
**Rating:** 3
**Confidence:** 3

**Summary:**

This paper reduces the storage requirements for federated unlearning using compression. For $U$ users, during FL, if the update from client $u\in [U]$ at each round $t$ is $h_t^{u}$, then the centralized model is updated as $w_{t+1} = w_t + \frac{1}{U}\sum_{u\in [U]} h_t^{u}$. Here, if $w_t^{u}$ is the model learned by client $u\in [T]$ at round $t$, $h_t^{u} = w_t^{u} - w_t$.

The goal of federated unlearning (FU) in the **train-from-scratch** (Liu 2023) model for a specific client $\tilde{u}\in [U]$ is to
remove all contributions of this client to the model. This would imply the following model $w_t^\star$ at every round $t$, where
$$
\begin{align*}
w_{t+1}^\star = w_t^\star + \frac{1}{U-1}\sum_{u\in [U], \tilde{u}\neq u} h_{t}^{\star, u},\quad h^{\star, u} = w_t^{\star, u} - w_t^\star \quad w_0^\star = w_0
\end{align*}
$$

Here, $w_t^{*, u}$ is the model update at client $u$ at round $t$ starting from $w_t^\star$. The goal is to modify $w_t$ to obtain $w_t'$ such that $\|w_t^\star - w_t'\|$ is small. One existing approach (Huynh et al 2024), uses skewed updates,
$$
\begin{align}
w_{t+1}' = w_{t+1} + \sum_{j=0}^t (1 + \alpha)^{t-j} \delta_j',\quad \delta_j' = \frac{1}{U(U-1)}\sum_{u\in [U], u \neq \tilde{u}} h_j^{u} - h_j^{\tilde{u}}
\end{align}
$$
Note that this approach, and several existing approaches for FU require storing the complete history of updates for all clients. This corresponds to $T\cdot U \cdot m \cdot b$ bits of storage, where $m$ is the number of parameters in model and $b$ is the number of bits per model parameter.

This paper proposes using dithered quantization, DQ and its unbiased variant SDQ, and sparsification techniques, topk and randk, to compress each model update. Then, in Eq (1), it proposes to use $Q(h_j^u) - Q(h_j^{\tilde{u}})$, where $Q(x)$ is the reconstructed version of a vector $x\in \mathbb{R}^m$ using a given compression scheme.



For skewness $\alpha=0$, step size $\eta_t$, only $1$ local step, bounded loss gradients by $M$, and unbiased quantization with variance of compression error $\mathbb{E}[\|Q(x) - x\|^2] = \sigma^2,\forall x$, the authors show that by running Eq (1) with $\alpha=0$, the obtained model's distance from the unlearned model converges to $G + \frac{\sigma^2}{(U-1)}(\nu^{-1} + \pi^2/6)$ for step size $\eta_t = 1/(t+\nu)^\lambda$ with $\nu >0, \lambda >1$. (Lemma 4.1, Theorems 4.3 - 4.4, Corollary 4.1).

Further, in Section 5, the authors experiment their methods against the baselines of full **train-from-scratch**, Eq (1) with no compression,different compressors and vanilla FL on NN with CIFAR10 and MNIST datasets with a malicious client. By using unlearning to learn the contributions of this malicious client, the authors show that increasing the rate of compressor results in better protection against the malicious client (Figure 3, Table 2).

**References**
- (Liu et al 2023) A survey on federated unlearning: Challenges, methods, and future directions. ACM Computing Surveys.
- (Huynh et al 2024) Fast-fedul: A training-free federated unlearning with provable skew resilience. ECML PKDD.

**Strengths:**

- **Important problem in federated unlearning**: In federated settings, number of users $U$ can be extremely large, and the number of rounds for an iterative optimization algorithm, $T$, can also be very large. Storage at the server does become a huge bottleneck even when we need to unlearn the contributions of just a single client.

- **Interesting experiments**: Unlearning the contributions of a malicious client is an interesting application of unlearning. Further, using the performance of model on the data of malicious client is a nice metric to study unlearning performance. Although this experiment has been proposed in (Huynh et al 2019), the authors use it to test performance of different compressors by varying the rates of these compressors.

**References**
- (Huynh et al 2024) Fast-fedul: A training-free federated unlearning with provable skew resilience. ECML PKDD.

**Weaknesses:**

- **Theory**:
    1. From AS1, number of local steps is $1$, which does not include FedAvg and from AS2, the function has bounded gradients. This makes the analysis easy, however, existing works in FL have relaxed these assumptions to much weaker ones(Stich and Karimireddy 2020). Further,the analysis local steps $>1$ for even federated unlearning has already been attempted by (Tao et al 2024) and it seems that it can be extended to this case.

    2. The unbiasedness and fixed variance of compression error in AS3 are extremely strong assumptions for a compressor and no proposed compressor satisfies these assumptions. Note that topk is a biased compressor, and randk and lattice quantization only satisfy these assumptions if the updates lie on a sphere or in the lattice support. Several weaker notions of compressors exist in literature (see Definitions 1,2,4 and 5 in (Beznosikov et al 2023)), which cover a large family of compressors including all the compressors stated in this paper.

    3. For (Huynh et al 2024), setting $\alpha=0$ (AS1) implies no skewness in updates. Skew estimation is an important contribution of (Huynh et al 2024) which necessitates $\alpha=0$. Can the authors provide problem settings, for instance loss functions, where using $\alpha=0$ is still valid?

    4. The choice of step size $\eta_t = \frac{1}{(t+\nu)^\lambda}, \lambda >1$ does not include the step size $\frac{1}{t + \nu}$. For bounded gradients, such a step size becomes essential to show convergence of final iterate of gradient descent( see Theorem 4 in (Koloskova et al 2019)), otherwise an averaged iterate is required (section 9.1 in (Garrigos &)). With $\lambda >1$, while the unlearning procedure might not diverge, but the actual FL algorithm might also not converge. Can the authors provide a reference for convergence of even GD with bounded gradients with the given step size?

- **Experiments**: In Figure 2, and Line 415, the SNR for a NN model is defined as $\log (Var(w_t')/Var(w_t' - w_t''))$. Note that NN weights can be permuted such that the weights differ but outputs of two networks remain the same. Therefore, using variance, via $\ell_2$ norm of model weights is not an appropriate metric. Further, can authors explain how this metric actually measures signal-to-noise ratio? If this metric has been used previously, they should cite it.



- **Comparison to Existing works**: (Jiang et al 2024) only stores model updates for clients and rounds which incur large updates. Therefore, without any compression, they reduce the storage dependence on $U$ and $T$. This is an important baseline both theoretically and for experiments to the proposed method.


**References**
- (Tao et al 2024) Communication efficient and provable federated unlearning. VLDB.
- (Beznosikov et al 2023) On Biased Compression for Distributed Learning. JMLR.
- (Stich and Karimireddy 2020) The error-feedback framework: Better rates for sgd with delayed gradients and compressed communication. JMLR.
- (Huynh et al 2024) Fast-fedul: A training-free federated unlearning with provable skew resilience. ECML PKDD.
- (Koloskova et al 2019) Decentralized Stochastic Optimization and Gossip Algorithms with Compressed Communication. ICML.
- (Garrigos & Gower 2023) Handbook of Convergence Theorems for (Stochastic) Gradient Methods. Arxiv.
- (Jiang et al 2024) Towards Efficient and Certified Recovery from
Poisoning Attacks in Federated Learning. Arxiv.

**Questions:**

- How does heterogeneity affect unlearning? The authors perform experiments for dirichlet class imbalance in Figure 5, but they provide no insights on it. Further, using real federated datasets
- Ideally, for any Federated unlearning algorithm, the authors can use compressed model updates instead of actual model updates. Assuming these are unbiased compressors, with small MSE,  are there existing Federated unlearning algorithms, like FedRecover, Crab or Verifi,  where using these compressors would fail? For instance, where higher order information is used, or MSE does not suffice?
- **Best compressors for Distributed Mean Estimation**: The model averaging step in FL is a form of distributed mean estimation with mean of updates to be estimated. For this problem, (Suresh et al 2017) propose the optimal quantizers. While the problem for unlearning is not mean estimation, rather that of differences $(Q(h^{u}) - Q(h^{\tilde{u}}))$, the optimal DME compressors instead of randk and topk might serve as a good starting point. Can the authors check if these compressors are actually covered by dithered lattice quantization?

**References**-
- (Suresh et al 2017) Distributed Mean Estimation with Limited Communication. ICML

---

> ### Author Response · Authors · 2024-11-20
>
> Thank you very much for the thorough review and insightful comments! We also appreciate the kind feedback regarding the studied problem and experiments!
> Before going into each comment, we would like to address your criticism regarding the simplicity of the assumption adopted in our theoretical evaluation in Subsection 4.2. To that aim, we wish to emphasize the following points:
> - **P1: Relevance of the setup**. We tackle a relevant problem in distributed machine learning. There is a need for federated unlearning, which directly follows from the growing popularity of federated learning (what if some user is detected at some point, perhaps even much after training is concluded, as being malicious, compromised, buggy, or just wishes to be forgotten?). Storage here becomes a real bottleneck, making one question whether this can even be done (as raised by Reviewer F5mS).
> - **P2: Novelty of the solution**. The solution we propose - use lossy compression when storing - has not yet been considered for federated unlearning. It is indeed simple, and based on established tools studied in other domains (including federated learning), but that is what makes it attractive and easy to apply.
> - **P3: Usefulness of our analysis**. The additional assumptions introduced when analyzing convergence are indeed restricted quite basic methods for learning, unlearning, and compressing. Since we are the first to look into server-side federated unlearning with lossy compression, it makes sense in our humble opinion to consider a basic learning-unlearning setup in our analysis, particularly given the scarcity of convergence characterizations for standard (non-compressed) server-side federated unlearning. Our characterization is expected to motivate future research on analyzing more involved and less restrictive scenarios, as was done in the domain of compressed federated learning.
>
> With the above in mind, we proceed to  address the comments and answer the questions in the same order in which they were indexed by the reviewer.
>
> - **W1**: Thank you for bringing this up. We are aware of both mentioned papers and indeed, federated learning and unlearning analysis exist in literature with relaxed assumptions. Nevertheless, we believe that the observation that our analysis is easy is in fact a good and desirable thing, particularly given the fact that our work is the first to study lossy compression for federated unlearning (P2). A simple and easy-to-follow analysis is not only accessible, but can motivate extensions to more involved settings (as noted in P3).
> Yet, even with these adopted assumptions, we managed to obtain surprising results. One of which is that our upper bound, on the difference between the desired model that is trained from scratch and the model unlearned from lossy compressed stored updates, outperforms the state-of-the-art known bounds for non-compressed decentralized server-side unlearning, even when lossy compression is incorporated. Our characterization also specializes a bound for non-compressed distributed unlearning. Before our work, it was not clear if such a bound can be even achieved. As an example, (Huynh et al 2024) only postulates that the growth rate of this difference (between non-compressed unlearning and train-from-scratch models) is at most exponential with the number of federated learning iterations.
> Our theoretical analysis may be an important first step toward a proof using relaxed assumptions, which, as the reviewer mentioned, could potentially be extended further. Following the reviewer’s comment, this would be discussed in the conclusions section of our revised paper.

---

> > ### Author Response · Authors · 2024-11-20
> >
> > - **W2**: Thanks for raising this important point. Indeed, there are weaker and more generic compared to the ones assumed in our analysis, that are possibly amenable to convergence studies. Nonetheless, in light of P1-P3, we do not believe that this diminishes our contribution, but rather indicates on its potential to motivate future research on this study, as was the case with, e.g., (Alistarh et al 2017), which adopted a similar lossy compression model and was highly influential in the domain of compressed federated learning.
> > Specifically, regarding the ability to assured that the model updates lie on a sphere or in the lattice support, this can be quite simply achieved via different techniques of overloading prevention, commonly applied in the quantization literature (Gray and Neuhoff 1998), (Widrow et al. 1996). For instance, in order to prevent overloading, namely, that each continuous-amplitude value lies in the quantizer support, the model updates vector h can be scaled by $\zeta || \boldsymbol{h} ||$ for some parameter $\zeta > 0$. This setting guarantees that the each L-sized sub-vector obtained reside inside the L-dimensional ball with radius $\zeta^{−1}$. Note that the scalar quantity $\zeta || \boldsymbol{h} ||$ depends on the vector $\boldsymbol{h}$ , and must thus be quantized with high resolution and conveyed to the server, to be accounted for in the decoding process. The overhead in accurately quantizing the single scalar quantity $\zeta || \boldsymbol{h} ||$  is typically negligible compared to the number of bits required to convey h, hardly affecting the overall quantization rate. In light of the reviewer’s comment, overloading prevention would be reviewed in our revised paper, right after the presentation of the DQ and SDQ compressors.
> >
> > - **W3**: We assume that you meant “...which necessitates $\alpha\neq 0$. Can the authors provide...where using $\alpha\neq 0$ is still valid?”. Please note that the algorithmic approach of (Huynh et al 2024) still holds under the specific setting of zero skewness. More specifically, right after equation (14) in (Huynh et al 2024), the authors constrain $\alpha \in [-\mathcal{K}$, $\mathcal{K}$] for  $0\leq\mathcal{K}$. Therefore, the proposed method of (Huynh et al 2024) is also applicable using $\alpha= 0$, in line with the setting of AS1 in our paper, for any problem settings and loss functions; while resulting with simplified and easy-to-follow analysis. Following the reviewer’s comment, in the revised version, this issue would be emphasized right after the introduction of AS1.
> >
> > - **W4**: We would like to note that the study of (Koloskova et al 2019) focuses on a serve-less framework, comprising merely the edge users; and thus, proving convergence of final iterate of gradient descent is required in such a case. In contrast, in our setting a server orchestrating the training does present. Consequently, by-definition, an averaged iterate exists in each global step, where all the users transmitted their updates to the server.
> > Commonly, (S)GD works indeed concern the learning rate scaling to be linear. Nevertheless, recent works, such that of (Balles et al 2024), reveals other useful alternatives for choosing an adaptive learning rate scaling for SGD while assuming gradient magnitude to be bounded; involving estimating the norm of the risk function’s gradient per round. While we are not aware of a work proving GD convergence using a polynomial step-size, i.e., $\eta_t = \frac{1}{(t+ν)^ \lambda} , \lambda > 1$; we are neither familiar with one contradicting its possible convergence. This, by itself, gives rise to a fascinating research question.
> > It is noted, though, that from a numerical point of view, the polynomial step size can be effectively simulated to be a linear one by taking $\lambda  \to 1$. Despite that, due to the reviewer comment, in the revised version we will refine Theorem 4.4. and Corollary 4.1. statements to be applicable, with the discussed learning rate, once the asymptotic convergence of the federated learning model is achievable.

---

> > > ### Author Response · Authors · 2024-11-20
> > >
> > > - **W5**: Using this SNR formulation is very common in lossy compression analysis (see Part V of (Polyanski and Wu 2014)), and is adopted in some key results in statistical quantization, e.g., the celebrated 6-dB rule of thumb. By choosing the described metric, we concern the signal as the input to the compression mechanism, or, alternatively, the non-compressed weights $\boldsymbol{w}'_t$, and the noise, in our case being the compressor error, as the difference between the quantizer’s input and output $\boldsymbol{w}'_t -\boldsymbol{w}''_t $. The variance is then numerically evaluated as the quantities are assumed to be stochastic.
> > > In practice, our numerical study quantitatively shows that under some assumptions convergence in the weights guarantees convergence in model performance. This follows since we first show that despite compression, we obtain close resemblance in weights (via the SNR metric); and then we show that it indeed leads to similar performance in terms of backdoor resiliency (via the main and attack tasks accuracy).
> > > According to the reviewer’s comment, in the revised version, we will justify the usage of the adopted SNR metric.
> > >
> > > - **W6**: We thank the reviewer for noting this important related work. We note that a similar approach to that (Jiang et al 2024), which samples subset of users for reducing storage footprint, is also employed in (Huynh et al 2024). Accordingly, as our experimental study also considers the unlearning procedure of (Huynh et al 2024), we already include such techniques in our numerical evaluation. Nevertheless, we view the integration of lossy compression as a complementary tool to those techniques, rather than a baseline. That is, regardless the amount of updates being stored, one would definitely benefit from storing each in a compressed form; having that compression not significantly deteriorate the unlearning performance. Yet, of course, is it of crucial importance once considerably large amount of updates are needed to be stored. Following the reviewer’s comment, the incorporating of lossy compression as a complementary tool into methods reducing the storage footprint of unlearning in an indirect manner, will be discussed in the conclusions section of the revised paper.
> > >
> > > - **Q1**: Please note that lines 802-809 in the submitted paper discuss the trends observed in Figure 5, and specifically the effect of heterogeneity on unlearning. The corresponding text is repeated below for ease of reference:
> > > *Next, in Fig. 5 we evaluate the performance of all baselines on the backdoor task test-set, with respect to varying values of β, which controls the degree of imbalance (lower values indicate greater imbalance), while keeping the compression rate R = 2.3 fixed. The top of the figure illustrates variations in the backdoor accuracy, which result from the fact that the number of backdoor samples is proportional to the attacker’s local dataset size. Adjusting β alters the number of samples allocated to each user, thereby influencing the backdoor attack performance on the vanilla FL model. As a result, we do not expect to observe consistent behavior across different values of β in terms of backdoor accuracy. However, we do observe persistent trends across all methods for a fixed β, keeping the best-to-worst performers order identical to that in Fig. 3.”*
> > > Finally, our numerical evaluations follows the framework and dataset covered in (Huynh et al 2024). Nevertheless, other federated datasets, and particularly real ones, are left as a main extension for future study.
> > >
> > > - **Q2**: Great question! we would like to clarify that further investigation is required to adequately answer this, necessitating thoroughly reading existing unlearning algorithms, as the ones mentioned by the reviewer; and figuring their implied encapsulated information needed. It is very interesting and an indication on the multitude of possible follow up studies which arises from our work.
> > >
> > > - **Q3**: Another great question! According to our humble understanding, the optimal DME compressors covered in (Suresh et al 2017), elaborated as stochastic uniform quantizers in section 2 therein, cannot be viewed as forms of the dithered lattice quantization adopted in our paper; even when the compressor dimension is 1. This is mainly because there, the quantized value for each coordinate is generated independently with private randomness, whereas in dithered quantization the dither is generated in an i.i.d fashion among all entries.
> > >
> > > References
> > > - (Huynh et al 2024) Fast-fedul: A training-free federated unlearning with provable skew resilience. ECML PKDD.
> > > - (Gray and Neuhoff 1998), Quantization. IEEE TIT.
> > > - (Widrow et al. 1996) Statistical theory of quantization. IEEE TIM.
> > > - (Balles et al 2024). On the Choice of Learning Rate for Local SGD. TMLR.
> > > - (Polyanski and Wu 2014) Lecture notes on information theory. ECE563 (University of Illinois Urbana-Champaign).

---

> > > > ### Author Response · Authors · 2024-11-22
> > > > **Thanks again for your review and time**
> > > >
> > > > Thanks again for your thoughtful review! We believe that we have addressed all of your concerns and questions in our response above. We would love to receive any additional feedback you may have. Do you have any follow-up questions? We are excited to engage in further discussions this week! Please let us know.
> > > >
> > > > Thank you very much, and we look forward to hearing from you.

---

> > > > > ### Comment · Reviewer_Pw5D · 2024-11-26
> > > > > **Response to Authors' rebuttal**
> > > > >
> > > > > I thank the authors for a detailed rebuttal. Some of my concerns have still not been addressed by the rebuttal.
> > > > >
> > > > > These are the **concerns that the rebuttal addressed**.
> > > > > 1. **Novelty**:   I am satisfied with the authors' response on this and I feel that part of their rebuttal in W1 should be included in the main paper to inform the reader that the simplifying assumptions are because this is the first analysis of even non-compressed distributed unlearning in the train-from-scratch case.
> > > > > 2. **Assumptions on compressors**: I am satisfied with the authors' explanation, however, I feel that using a more generic compressor definition, for instance with bounded variance instead of constant or those mentioned in my review, might have been easier to analyze theoretically, as some of them have been designed to make convergence analysis of distributed learning easier. This would have increased the contribution of the paper, however, I'm satisfied with the authors' explanation and using the compressors proposed in their paper as a starting point.
> > > > >
> > > > >
> > > > >  The **concerns that the rebuttal did not address** are as follows.
> > > > >  1. **Choice of step size**: From Theorem 3.2.1 in (Nesterov 2018), the minimax lower bound for convex Lipschitz functions on a compact bounded domain is $\Omega(\frac{1}{\sqrt{T}})$ . From Theorem 3.2.2 and algorithm in (3.2.14) in (Nesterov 2018), setting the step size to be $\frac{1}{\sqrt{t}}$, gradient descent achieves the lower bound.  Further, in algorithm (3.2.14), the sum of step sizes diverging is an important criterion, otherwise, the algorithm does not make much progress from the optima.
> > > > > The step size presented in this paper is $1/(t+\nu)^\lambda$ with $\lambda >1$, so its sum and the sum of its square converge. Consider the upper bound in Theorem 3.2.2, where we approximate the sum of step sizes by their integral, from iteration $0$ to $T-1$. Then, $\sum_{t=0}^{T-1} \eta_t \approx \frac{1}{\lambda - 1}\left(\frac{1}{\nu^{\lambda-1}} - \frac{1}{(T-1+\nu)^{\lambda-1}}\right) \geq \frac{1}{\lambda - 1}(\frac{1}{\nu^{\lambda - 1}} - \frac{1}{(1 + \nu)^{\lambda - 1}}) = C$ for some constant $C$. The last inequality is obtained by using $T - 1 \geq 1$. Now, plugging this into (Theorem 3.2.2) in (Nesterov 2018), the first term in the upper bound on RHS is, $\frac{MR^2}{C}$. Note that the LHS is the suboptimality in function value of the $T^{th}$ iterate, but the suboptimality in function value of the initial iterate is $MR^2$. Therefore, for the step size proposed in this paper, the suboptimality of any iterate is bounded by a constant times the initial suboptimality, and is therefore not decreasing, which is required for convergence. The reason why this happens lies in the sum of step sizes diverging, as if they don't diverge, then the final iterate is not far away from the initial iterate and can thus be arbitrarily far away from the actual minima. **This is a fundamental problem in the proposed method, as unlearning requires sum of step sizes to be converging while learning requires sum of step sizes to be diverging**. So, at least based on the current method, both learning and unlearning cannot happen at the same step size. Specifically, the step size proposed by authors can be used for unlearning but it implies that the model using this step size for learning would not converge to its optimal.
> > > > >
> > > > > 2. **SNR**: Formulating the SNR as ratio of variances is a valid metric for linear or generalized linear models, however, not for multi-layer neural networks.  In experiments, the authors use multiple layer NNs. This is because the $\ell_2$ distance is not a valid distance metric for multiple layer neural networks due to permutation invariance. Consider a neural network of 2 fully connected layers. If I interchange the weights of the first and second hidden node in first layer, then I can interchange the weights of all nodes in the second layer connecting to first and second hidden node in the first layer, such that the output of this network and the original network is exactly the same. However, the $\ell_2$ distance between the weight matrices of these two networks is non-zero. As the authors use multiple layer NNs for their experiments, they should not compute SNR in terms of $\ell_2$ variance of weights, as it might be erroneous.
> > > > >
> > > > > For all the remaining issues in my review, either the authors' rebuttal has been more or less satisfactory or the issue was not very significant. I think the choice of step size is a key problem in the paper, as it appears that the authors require converging step sizes for unlearning, but learning requires diverging step sizes. So, the authors need either a step size or a problem setting or both, such that their proposed step size works for both learning and unlearning in the problem setting.
> > > > >
> > > > >
> > > > >
> > > > > **References**
> > > > > - (Nesterov 2018) Lectures on Convex Optimization. Springer.

---

> ### Author Response · Authors · 2024-11-27
>
> Dear Reviewer Pw5D,
>
> Thank you very much for your detailed and thorough explanation. We now believe we have a clear understanding of how to reformulate the setting, particularly in determining appropriate learning rates that ensure the convergence of both the learning and unlearning algorithms. Consequently, we would like to withdraw our submission for further investigation and improvement.
>
> Once again, thank you for your insightful and didactic review.
>
> Best regards,
>
> The Authors

---

### Comment · Area_Chair_LiHX · 2024-11-26
**Response**

Dear Reviewers,

The authors have provided their rebuttal to your questions/comments. It will be very helpful if you can take a look at their responses and provide any further comments/updated review, if you have not already done so.

Thanks!

---

### Note · Authors · 2024-11-27

I have read and agree with the venue's withdrawal policy on behalf of myself and my co-authors.